# A Simulation Study of Tolerance of Breathing Amplitude Variations in Radiotherapy of Lung Cancer Using 4DCT and Time-Resolved 4DMRI

**DOI:** 10.3390/jcm11247390

**Published:** 2022-12-13

**Authors:** Guang Li, Admir Sehovic, Lee Xu, Pawas Shukla, Lei Zhang, Ying Zhou, Ping Wang, Abraham Wu, Andreas Rimner, Pengpeng Zhang

**Affiliations:** 1Department of Medical Physics, Memorial Sloan Kettering Cancer Center, New York, NY 10065, USA; 2Department of Radiation Oncology, Memorial Sloan Kettering Cancer Center, New York, NY 10065, USA

**Keywords:** time-resolved 4DMRI motion simulation, 4DCT motion simulation, motion-incorporated radiotherapy planning, motion-tolerance of ITV-based planning, volumetric-modulated arc therapy (VMAT), stereotactic body radiotherapy (SBRT), respiratory motion irregularities, lung, liver, pancreatic cancer

## Abstract

As patient breathing irregularities can introduce a large uncertainty in targeting the internal tumor volume (ITV) of lung cancer patients, and thereby affect treatment quality, this study evaluates dose tolerance of tumor motion amplitude variations in ITV-based volumetric modulated arc therapy (VMAT). A motion-incorporated planning technique was employed to simulate treatment delivery of 10 lung cancer patients’ clinical VMAT plans using original and three scaling-up (by 0.5, 1.0, and 2.0 cm) motion waveforms from single-breath four-dimensional computed tomography (4DCT) and multi-breath time-resolved 4D magnetic resonance imaging (TR-4DMRI). The planning tumor volume (PTV = ITV + 5 mm margin) dose coverage (PTV D95%) was evaluated. The repeated waveforms were used to move the isocenter in sync with the clinical leaf motion and gantry rotation. The continuous VMAT arcs were broken down into many static beam fields at the control points (2°-interval) and the composite plan represented the motion-incorporated VMAT plan. Eight motion-incorporated plans per patient were simulated and the plan with the native 4DCT waveform was used as a control. The first (D95% ≤ 95%) and second (D95% ≤ 90%) plan breaching points due to motion amplitude increase were identified and analyzed. The PTV D95% in the motion-incorporated plans was 99.4 ± 1.0% using 4DCT, closely agreeing with the corresponding ITV-based VMAT plan (PTV D95% = 100%). Tumor motion irregularities were observed in TR-4DMRI and triggered D95% ≤ 95% in one case. For small tumors, 4 mm extra motion triggered D95% ≤ 95%, and 6–8 mm triggered D95% ≤ 90%. For large tumors, 14 mm and 21 mm extra motions triggered the first and second breaching points, respectively. This study has demonstrated that PTV D95% breaching points may occur for small tumors during treatment delivery. Clinically, it is important to monitor and avoid systematic motion increase, including baseline drift, and large random motion spikes through threshold-based beam gating.

## 1. Introduction

In radiotherapy, a mobile tumor driven by respiration is usually treated by targeting the entire tumor motion trajectory, namely the internal tumor volume (ITV), which is delineated on respiratory-correlated (RC) 4-dimensional computed tomography (4DCT), as the standard of care. As the ITV is defined as the union of all clinical tumor volumes (CTV) within 4DCT, ITV-based planning assumes that the tumor stays within the ITV most time during treatment while slight motion variations can be compensated by the natural dose-falloff penumbra around the planning tumor volume (PTV = ITV + setup safety margin) [1,2]. However, several recent clinical observations using X-ray fluoroscopic imaging during the treatment of 37 lung and liver patients [3] and dynamic 2D cine magnetic resonance imaging (MRI) during simulation [4] or treatment [5] have shown that inter- and intra-fractional mean tumor motion variations can be substantial (as large as tripled motion range of the 4DCT) due to patient’s breathing irregularities. Beyond common breathing variations, lung patients may also suffer from pain or lack of comfortability, which varies daily or even during treatment, and such discomfort may lead to a change in the breathing pattern, therefore, causing ITV variations. Moreover, the ITV delineated from the snapshot composite single-cycle 4DCT simulation may not represent the target mobility accurately for a treatment fraction over 20 min during a multi-fractional radiotherapy treatment [5]. As a result, the treatment may deviate from what is planned, potentially leading to a sub-optimal treatment [3,4,6,7]. So far, it remains a challenging question about how much motion variation can be tolerated clinically during plan delivery, and therefore it is necessary to investigate this subject quantitatively to guide clinical practice.

Respiratory-induced tumor motion variability has long been recognized as uncertainty in the radiotherapy of a mobile tumor [8,9], and has been studied using various 2D imaging techniques in real-time, including kilovoltage (kV) fluoroscopy, megavoltage (MV) treatment beam imaging, and more recently dynamic 2D cine MRI [4,10]. Using CyberKnife imaging data from 37 lung patient treatments it was found that intra-fractional tumor motion had ~60% reproducibility while the combined intra- and inter-fractional motion had only ~30% reproducibility [11]. Using fluoroscopic imaging, a study on long- and short-term tumor motion variations in 37 lung and liver patients found that half of the tumors had significant motion amplitude variations (>5 mm) [3]. Using dynamic 2D cine MRI in an MR-integrated linear accelerator, another study with 5 lung cancer patients found the ITV area from 10-s scans only accounts for 46% of the ITV from 20-min scans in 15 fractions, suggesting the ITV delineated from 4DCT based on a composite single-cycle may not be statistically reliable [4], especially for hypo-fractional stereotactic body radiotherapy (SBRT). Recently, time-resolved (TR) 4DMRI has been developed to capture multi-breath volumetric motion images on the fly [12,13,14], so that the motion variability of a tumor and organs at risk (OARs) over multiple breathing cycles can be imaged and available to retrospectively assess patients’ motion variability.

To account for tumor motion in radiotherapy planning, the concept of the probability density function (PDF) of tumor motion was first introduced to incorporate motion through a convoluted dose calculation [15,16,17]. However, studies have shown that motion PDFs may not be reproducible, including a large difference between a 5-s scan and a 5-min scan [18], a high variance of ITV overlap values from multiple 10-s scans [5], and a strong dependency on native tumor motion range [19]. Recently, a motion-incorporated VMAT planning technique has been reported to incorporate motion in VMAT planning to assess plan delivery quality in the presence of random motion in prostate cancer [20] and respiratory tumor motion in lung cancer [21] using an existing treatment planning system (TPS, Eclipse, version 16.1, Varian Medical Systems, Palo Alto, CA, USA) based on in-house developed plug-in scripts. This method introduces tumor motion waveform as the isocenter motion profile while keeping the clinical beam with optimized multi-leaf collimator (MLC) motion unchanged, except for segmenting the arc beam(s) into many static beam fields at the VMAT-plan control points. Therefore, the dosimetry consequence of tumor motion variations can be simulated and assessed using TR-4DMRI motion waveforms at clinically acceptable motion detection accuracy (<2 mm).

In this study, we applied the motion-incorporated planning strategy by altering the treatment isocenter using the tumor motion waveforms extracted from 4DCT and TR-4DMRI. Clinical ITV-based VMAT plans with optimized MLC motions of 10 lung cancer patients were used for the motion-incorporated planning study under an IRB-approved protocol. As a control experiment, the equivalency of the motion-incorporated plans based on 4DCT to the ITV-based VMAT plans was first assessed. Namely, the dosimetry of the motion-incorporated plan and ITV-based plan should be identical with a negligible difference. TR-4DMRI images over 120 s were applied to introduce patient-specific multi-breath motion variations, which deviated from 4DCT, in the motion-incorporated plans. In addition to these native tumor motion waveforms from 4DCT and TR-4DMRI, six extra motion waveforms were introduced using linear scaling-ups by 5, 10, and 20 mm. So, eight motion-incorporated plans per patient were created to simulate motion amplitude variation during treatment delivery to evaluate the PTV coverage and the additional motions beyond 4DCT and TR-4DMRI native motions that cause the first and second breaching points at D95% ≤ 95% and D95% ≤ 90%, respectively.

## 2. Materials and Methods

### 2.1. Tumor Motion Waveform from Time-Resolved 4DMRI Scans at Simulation

Under an IRB-approved protocol, ten lung cancer patients were scanned for TR-4DMRI within 3 h after clinical 4DCT simulation using the same body immobilization device with both arms up in a 3T MR scanner (Ingenia v5.0, Philips Healthcare, Amsterdam, The Netherlands). To reconstruct super-resolution TR-4DMRI, two types of T1w 3D cine images were acquired per subject, including high-resolution (2 × 2 × 2 mm^3^) images in 20-s breath-hold (BH) and low-resolution (5 × 5 × 5 mm^3^) image at 2 Hz in three 40-s free-breathing (FB) scans. A turbo fast-gradient echo pulse sequence and 3D cine acquisition were applied, with the parallel imaging acceleration (SENSE, a factor of 6 for FB and 2.5 for BH), partial Fourier transformation (0.75), and the CENTRA acquisition order [14]. The TE/TR of 1.9/4.2 ms, flip-angle of 15°, and bandwidth of 4000 Hz were applied. The readout direction was set to the superior-to-inferior direction, the slice-encoding set to the anterior-to-posterior (coronal) direction, and the phase encoding set to the right-to-left direction. The same FOV (about 350 × 350 × 350 mm^3^), which covered the entire lungs and liver in the abdomen, was used for BH and FB MR image acquisitions.

TR-4DMRI images were reconstructed using a super-resolution method via deformable image registration (DIR) [14,22,23,24]. Using the displacement vector field (DVF) of DIR from the high-resolution BH image to the low-resolution FB images, the center of mass (COM) of the gross tumor volume (GTV) as a function of time (2 Hz) was used to create the tumor motion waveforms. The tumor motion waveforms were confirmed by extracting the diaphragm motion waveform and scaling it down linearly to the tumor motion range on every point in the waveform using an in-house program in MatLab (version 2020b, Natick, MA, USA), assuming a high correlation between the diaphragm motion and tumor motion. The native tumor motion waveforms from three 40-s scans were concatenated as the isocenter motion curve. To simulate motion amplitude increase, three additional motion waveforms were synthesized by linearly enlarging the native motion ranges by an extra 5, 10, and 20 mm as additional isocenter motion curves.

### 2.2. Clinical ITV-Based VMAT Planning and 4DCT Tumor Motion Waveforms

Clinical ITV-based VMAT plans of the 10 patients (3 hypo-fractional and 7 conventional) based on 4DCT images on a big-bore CT simulator (64-slice Brilliance, Philips Healthcare, Amsterdam, The Netherlands) were used in this simulation study. Each VMAT plan contained 2–4 arcs targeting the PTV, depending on the size and shape of the PTV, using the Eclipse planning system (Varian Oncology Systems, Palo Alto, CA, USA). The GTV was contoured in all 10 images within the 4DCT using the MIM Maestro (version 6.8, MIM Software, Inc, Cleveland, OH, USA), the ITV was created as the union of all GTVs with a margin of either 2 mm (early-stage) or 7 mm (mid/late-stage) to account for microscopic cancerous extension, and the PTV had a 5.0 mm isotropic expansion of the ITV. The PTV dose coverage was normalized to D95% = 100% as prescribed based on the dose-volume histogram (DVH) in the clinical VMAT plan, namely, at least 95% of the PTV volume received 100% prescribed dose.

The tumor motion waveforms were created by tracking the COM of the GTV within the one breathing cycle of 4DCT and concatenated repeatedly to contain multi-breaths. In addition to the native motion waveform (concatenated with replicates), three amplitude-enlarged waveforms were simulated by linearly scaling up the motion ranges with extra 5.0, 10.0, and 20.0 mm.

### 2.3. Motion-Incorporated VMAT Plan with a Moving Isocenter

Using two script programs (in C#) as plug-ins to the Eclipse planning system [20,21], the VMAT plans were first converted into composite plans with many static-beam fields at the control points (with a 2° interval) for all arcs, and then the plan isocenters were displaced based on the tumor motion waveforms while the original MLC motion and gantry rotation were applied to mimic VMAT plan delivery. A composite plan with static beams in all arcs was created as the motion-incorporated VMAT plan. The 8 waveforms derived from 4DCT and TR-4DMRI per patient were applied to move the isocenter to create 8 motion-incorporated plans. Therefore, the tumor motions were incorporated into the VMAT plans, reflecting dynamic tumor motion. It is worthwhile to note that this was a simplified approach to estimate the delivered dose to a rigid target as the respiratory-induced deformation of the tumor and OARs was ignored. For lung cancer patients, this rigid tumor assumption was reasonable as previously reported [25,26], and the dose variations to the OARs were out of the scope of this study.

The PTV dose coverage was evaluated using the D95% value: D95% = 100% was used to normalize all clinical plans and D95% ≤ 95% and D95% ≤ 90% were used as the first and second dose breaching points for the motion-incorporated plans using actual and scaled-up hypothetical tumor motion waveforms. The motion-incorporated plan quality (PTV D95%) was assessed as a function of tumor sizes and motion variations.

### 2.4. D95% Equivalency between Motion-Incorporated and ITV Plans with the Same 4DCT Motion

The motion-incorporated plan employed the moving isocenter approach, namely using the 4DCT tumor motion waveform to move the isocenter with the same leaf sequences, gantry arc rotations, and dose-calculation grid size (1.25 mm for SBRT plans and 2.5 mm for conventional plans) of the clinical ITV-based VMAT plans using the Eclipse TPS. This was a dynamic approach to handling tumor motion, while clinical ITV-based planning was a static approach. Presumably, the radiation dose distributions for the simulated (dynamic) plan and clinical ITV-based (static) plan should be equivalent as the two plans had the same moving target. To validate the hypothesis, we compared the two plans to illustrate the equivalency of the dynamic and static approaches in the control experiment. The PTV D95% values of the motion-incorporated plans were assessed and compared to the clinical ITV-based plans (D95% = 100%). The D95% comparison served as a control experiment to assess the uncertainties of the motion-incorporated plans based on the plan D95% differences. The minor D95% difference in the motion-incorporated control plan was used as a correction factor (to D95% = 100%) and applied to the other 7 motion-incorporated plans of the same patient.

### 2.5. Evaluation of PTV D95% Breaching Points in the Motion-Incorporated Plans

Using the original and scaled-up 4DCT and TR-4DMRI tumor motion waveforms, eight motion-incorporated plans per patient were built. For both 4DCT and TR-4DMRI motion-incorporated plans, the D95% values were obtained and plotted, illustrating the trend of the D95% decrease as a function of increased motion amplitudes. Linear and quadratic fittings of D95% vs. added motion amplitudes were performed to identify the corresponding extra motions at the first (D95% = 95%) and second (D95% = 90%) plan breaching points. Figure 1 illustrates the workflow of this study.

The motion-incorporated plan was built as a composite plan from all static plans at the control points of the clinical VMAT plan, as discussed in Section 2.3. Based on the plan comparison in the control experiment using 4DCT, a small correcting factor (the actual D95% value) was applied to normalize the other 7 motion-incorporated plans, as discussed in Section 2.4. The PTV D95% values were obtained through the process and used to assess motion variations that caused the first and second plan breaching points.

## 3. Results

### 3.1. Characterization of Lung Tumors and Their Original and Scaled-Up Motion Amplitudes

Table 1 tabulates the patient information, including the tumor locations, volumes, and native motions found from single-breath 4DCT and multi-breath TR-4DMRI. In this ten-patient pool, only patients 1 and 2 have native tumor motions around 1 cm (Figure 2), while the other patients have a smaller tumor movement due to patient-specific respiratory motion and tumor location. Three patients (2, 8, and 9) have a small tumor (GTV < 25 cm^3^, or ø < 3 cm in diameter), so they received hypo-fractionated stereotactic body radiotherapy (SBRT) treatment (Table 2). Figure 2 illustrates two examples of native and scaled-up motions (linearly by 5, 10, and 20 mm) based on the simulation of 4DCT and TR-4DMRI motion waveforms. Patient 1 is an irregular breather while patient 2 is a regular breather.

### 3.2. Comparison of Clinical ITV-Based Plan and Motion-Incorporated Plan as a Control

The average difference in PTV coverage is 0.6% between the motion-incorporated plan (D95% = 99.4% ± 1.0%) and clinical plan (D95% = 100%) with the original tumor motion extracted from 4DCT, as shown in Table 2. This illustrates the PTV dose equivalency between the dynamic and static planning approaches. Furthermore, the minor difference in PTV D95% between the two plans is used to correct the other motion-incorporated plans of the same patient before comparison analysis. The motion-incorporated plans using native motion waveforms from TR-4DMRI are also shown in Table 2. For the small pool of patients, limited breathing irregularities were observed in TR-4DMRI (Figure 2) so the plans are similar to the clinical plans, except for patient 8 whose motion-incorporated plan using native TR-4DMRI motion waveform results in PTV D95% = 94%.

### 3.3. Plan Quality Reduction with Scaled-Up Tumor Motion Amplitude from 4DCT and TR-4DMRI

A general trend of plan quality deterioration due to increased tumor motion amplitudes is shown in Table 2. Figure 3 illustrates 4 typical cases of PTV D95% reduction as a function of the added tumor motion. The dose breaching points (D95% = 95% and 90%) at a specified motion are shown. As quadratic fitting provides a better fit (with a significantly higher R^2^) than a linear fit, PTV D95% breaching points are determined based on the quadratic fits. Table 3 shows the first and second breaching points and the quadratic fitting R^2^ values. An extra motion of 4 mm on average is needed for D95% ≤ 95% in small-tumor SBRT treatments. For D95% ≤ 90%, more than half of the patients require a slight exploration, beyond the 20 mm range, such as patients 3, 4, 5, 6, 7, and 10.

### 3.4. PTV D95% Breaching Point as a Function of Tumor Sizes

In addition to the motion variations, the GTV size is also a factor that affects the ITV and PTV coverage as both factors affect the treatment targeting volume. Figure 4 illustrates that the first (D95% ≤ 95%) and second (D95% ≤ 90%) PTV breaching points as the function of GTV and ITV volume, suggesting that the smaller the tumor the more sensitive the plan quality (D95%) to the enlarged motion variation beyond the planned tumor motion based on the 4DCT simulation. As a comparison, TR-4DMRI adds more motion variations, which leads to larger variations (smaller R^2^) of both the first and second PTV D95 breaching points. As the tumor size increases, both the first and second breaching points tend to approach a level-off plateau, and the logarithm fitting has a much higher R^2^ than the linear fitting. Figure 4 shows the level-off trend as the GTV/ITV size increases.

## 4. Discussion

### 4.1. Radiation Penumbra and Tumor Motion Variability in VMAT Plan Delivery

Among the 10 patients’ VMAT plans, 3 of them are SBRT VMAT (patients #2, #8, and #9) with a smaller GTV-to-CTV margin (2 mm), while the others have a larger margin (7 mm) for conventional fractionation VMAT treatments. Although the dose falloff may be sharper in SBRT VMAT plans than in regular VMAT plans, the plan penumbra is universally present and not substantially different. In fact, the dosimetrists in our institution are always trying to achieve high PTV conformality with minimal dose spread and the major difference is the dose-calculation grid size (1.25 mm for SBRT VMAT and 2.5 mm for conventional VMAT). Therefore, in the comparison study, no differentiation is made in the planning techniques used for different prescriptions.

From previous reports, it is known that the PTV penumbra, or the dose spilling outside of the PTV, falling from the prescribed dose (100%) to field edges (50%), can in fact cover some tumor motion variations greater than the specified motion in 4DCT simulation [2,25]. However, it is unclear how much motion variations the PTV penumbra in VMAT plans can cover the PTV in treatment delivery in the presence of motion variations. This study quantifies tolerable motion variations in amplitude before reaching a clinically unacceptable PTV D95% value, denoted as the plan breaching point. It is worthwhile to note that the motion amplitude variations simulated in this study are uniformly and linearly enlarged motion ranges throughout the beam delivery time. Therefore, the simulated motions may represent the worst-case scenario as respiratory motion irregularities may not follow a uniform distribution for a long period during treatment. However, if the snapshot of motion simulation during 4DCT happens to be smaller than usual, then the tumor motions at treatment could be much greater, leading to a dose breaching point. Using TR-4DMRI simulation, breath-to-breath variations over a much longer time frame can be captured, and therefore it can help to determine a more representative tumor motion range for ITV delineation. In several recent studies, it has been demonstrated that the mean tumor motion during treatment can be as large as 2–3 times the motion from simulation 4DCT [3,4,5]. Therefore, the results from this study provide quantifications of plan quality and its degradation due to enlarged tumor motion. It is worthwhile to mention that the motion-incorporated planning in this study is a dynamic approach to include tumor motion and motion variation into treatment planning to simulate what may happen during treatment delivery. Therefore, the dosimetry comparison between the static (ITV-based) approach and dynamic (motion-incorporated) approach in tumor motion handling can reflect the difference between what is planned and what is delivered. To our best knowledge, this is the first time to report the quantified relationship between motion variations and VMAT plan breaching points using the motion-incorporated planning technique.

### 4.2. The Tolerable Motion Variation Thresholds in Clinical VMAT Treatments

The tolerable motion variations are a function of GTV and ITV size, as shown in Table 1 and Table 3 and Figure 4. For GTV smaller than 25 cm^3^ and ITV smaller than 50 cm^3^ regardless of their original motion, 5 mm additional motion can reduce the PTV D95% from 100% to 88–95%. With the increase of the GTV (from 30 to 140 cm^3^) and ITV (from 80 to 450 cm^3^), even a 10 mm addition to the original motions may not reduce the PTV D95% lower than 95%. As a matter of fact, the dependency on the tumor size seems diminished for GTV > 30 cm^3^ or ITV > 80 cm^3^, as shown in Figure 4. In contrast, a 4 mm extra motion on average will lead small tumors to reach their first breaching point and a 10 mm increase in motion decreases the PTV D95% from 100% to 85% for the smaller tumors. In addition to the tumor size, it is reasonable to expect that tumor shape and 3D motion beyond super-infer motion may also play a role, but they should be secondary factors and will be investigated in the future.

In general, the native tumor motion from this patient group is small (3 ± 3 mm) with the largest tumor motion range of 8.5 mm from the 4DCT simulation and 10.8 mm from TR-4DDMRI. Therefore, the ITV size increase from the GTV due to the small motions is not as dramatic as it would be otherwise. However, when evaluating ITV, the GTV size and motion range are combined, and therefore, the smaller tumor motion in this patient pool should not alter the conclusion of this study. For small tumors, when additional motion is added to simulate motion amplitude variation during treatment delivery, the actual ITV size increases are substantial relative to their GTV size due to the cubical relationship between the tumor diameter and the volume of the tumor motion envelope. On the other hand, for larger tumors, the relative increase of the actual ITV size would not be as significant as that of smaller tumors, and therefore, the added tumor motion may only cause a relatively smaller ITV increase, leading to a smaller impact to the plan quality (PTV D95%).

In addition to the target volume changes, dosimetrically the original tumor motion has been included in the control plan (the dynamic version of the ITV-based plan) regardless of its motion amplitude, demonstrating that the motion-incorporated VMAT plan (PTV D95% = 99.4% ± 1.0%) has the dose-equivalency (<1%) to the corresponding clinical VMAT plan (PTV D95% = 100%). This small D95% difference is corrected for the motion-enlarged and TR-4DMRI-based motion-incorporated plans of the same patient. Therefore, the bias from this motion-incorporated planning strategy is largely removed. In addition, the native tumor motion in this pool of patients is relatively small, for larger native tumor motions, the conclusion drawn here would still apply but might have a slightly smaller impact as the relative ITV increase is smaller. For tumors with large native motions, however, it has been reported that they tend to cause higher motion variations [19].

### 4.3. Motion Variations from Multi-Breath TR-4DMRI and Dosimetry Consequences

In this study, multi-breath TR-4DMRI data are applied for the first time to introduce native patient breathing irregularities into a simulation study to assess the quality of ITV-based plan delivery. Both similar and different tumor motion waveforms are observed in the multi-breath tumor motion waveforms in TR-4DMRI, deviating from the single-breath 4DCT motion waveforms, as shown in Figure 2. The dosimetry consequences of the motion amplitude variations are patient-specific: the PTV dose diversity is increased in some cases, but reduced in others, as shown in Figure 3 and Table 2 and Table 3. Therefore, compared with the 4DCT-based plans, TR-4DMRI introduces variations making the relationship of breaching points to GTV/ITV (Figure 4) less reliable (with a smaller R^2^). For the smaller tumors treated with SBRT, one out of three patients’ motion-incorporated plan reaches the first breaching point by native tumor motion observed in the TR-4DMRI. On average, 4 mm of extra motion on TR-4DMRI triggers the first breaching point, similar to that on 4DCT, but with a much-increased standard deviation, suggesting the dosimetry variability from tumor motion variability. In comparison with 4DCT results, the use of TR-4DMRI native and scaled-up motion data increases the variations with reduced R^2^ values to the relationship of the breathing points vs. GTV and ITV sizes, as shown in Figure 4.

The delivered radiation dose to the target can also be accessed based on real-time 4DMRI images acquired during treatment [12,13] and mapped with CT image intensity for frame-by-frame dose calculation [27]. However, although this might be the most accurate approach to incorporate tumor motion variations during treatment to assess the delivered dose, it is very time-consuming even for research purposes, and may not be very practical in the clinic. On the contrary, with the approximation of rigid tumors [26] in the motion-incorporated planning approach, the workload is much smaller and can be acceptable to analyze the delivered radiation dose in the presence of tumor motion variations.

### 4.4. Clinical Concerns on Tumor Motion Management

The results from this study suggest that extra caution should be taken for treating lung, liver, or pancreatic cancer patients with small GTVs (<25 cm^3^), as even a <5 mm motion amplitude increase would result in the first dosimetry breaching point in the delivery of a VMAT plan due to breathing irregularities, as shown in the multi-breath TR-4DMRI motion simulation. Thus, it is important to be cautious in hypo-fractional SBRT treatments, which have 5 fractions or fewer, whereby the averaging effect of having multiple fractions is lost in comparison with conventional fractionation. In the current radiotherapy clinic, the ITV-based VMAT treatment does not give special consideration to SBRT cancer patients, even though large motion variations have been reported [3,4], because no quantitative data are available to guide the treatment. Based on this study, the following actions would be considered beneficial to avoid possible sub-optimal treatments. First, the external motion surrogate used for 4DCT simulation, such as RPM (real-time positioning management, Varian Medical), should be used to establish a rough correlative external-internal motion relationship and applied during the treatment. If an MRI simulator is available in a clinic, multi-breath TR-4DMRI (or 2DMR cine) motion simulation is recommended [12,13,14], so the external-internal motion correlation can be better quantified as a more reliable representation of tumor motion [28], in contrast to the composite tumor motion within a single breathing cycle. Second, during treatment, if tumor motion is higher than that of simulation by 5 mm, the SBRT beam should be temporarily turned off (beam gating on thresholds) to avoid temporarily partial missing of the target until the breathing trace falls back to its simulation motion range. Therefore, the treatment quality can be ensured to deliver sufficient dose coverage to the PTV. Smaller than simulated motion range should not reduce the PTV dose coverage but may lead to increased dose to nearby OARs, which is out of the scope of this study, but is the focus of an ongoing investigation.

## 5. Conclusions

This study demonstrated that tumor motion variability due to patient breathing irregularities can cause substantial PTV coverage reduction indicated by the D95% value decreasing from 100% to 95% or 90%. For small tumors (GTV < 25 cm^3^ and ITV < 50 cm^3^) treated with hypo-fractional SBRT, VMAT plan delivery is vulnerable to a 5 mm additional motion and needs special attention to ensure treatment quality. Therefore, it should be cautious to use the ITV-based SBRT to treat small tumors without any online monitoring methods.

## Figures and Tables

**Figure 1 jcm-11-07390-f001:**
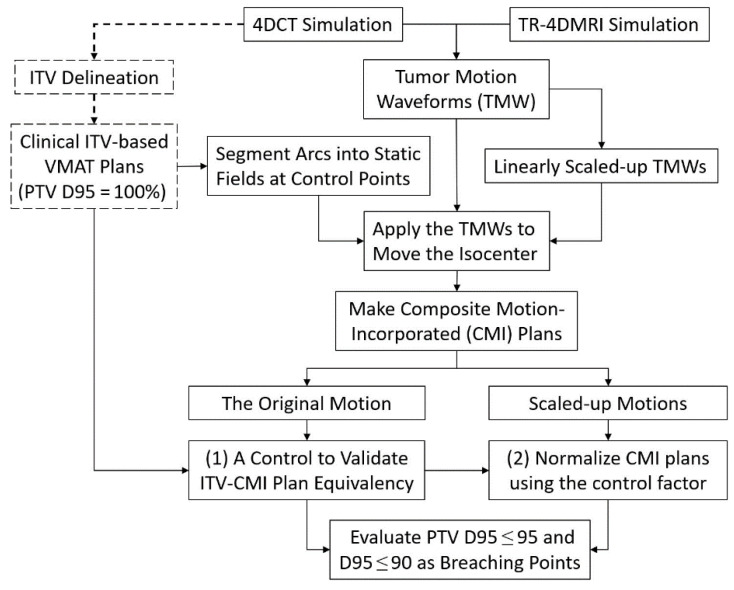
The flowchart of the motion-incorporated planning study. The dashed-lined sections are clinical VMAT planning, while the solid sections are performed in this study. The PTV D95 is used to compare the plan quality equivalency between ITV-based clinical VMAT plans and the motion-incorporated study plan with original tumor motion from 4DCT. The motions that trigger the PTV dose breaching points (D95 = 95% and 90%) are analyzed and reported.

**Figure 2 jcm-11-07390-f002:**
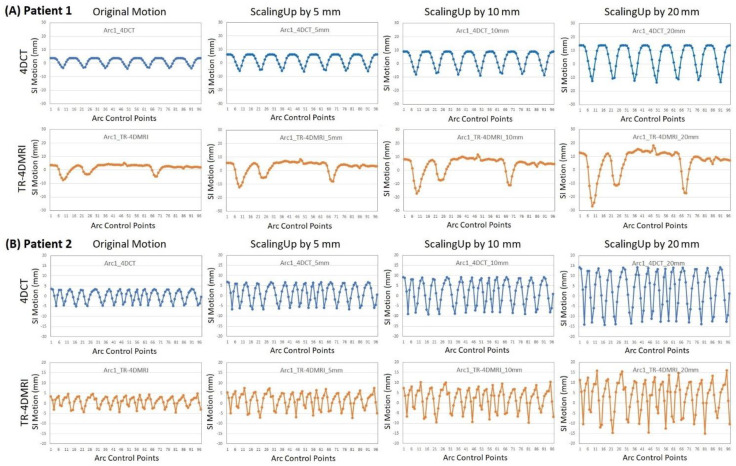
Examples of original and scaled-up tumor motion waveforms from 4DCT and TR-4DMRI simulation scans of two lung cancer patients. Note: exhalation has a positive motion while inhalation has a negative motion. Also, the *x*-axis is the control point (2° interval) in the arcs, so it is dependent on gantry speed. During the multi-breath TR-4DMRI scans, patient 1 (**A**) shows substantial tumor motion irregularities, while patient 2 (**B**) breathes rather regularly.

**Figure 3 jcm-11-07390-f003:**
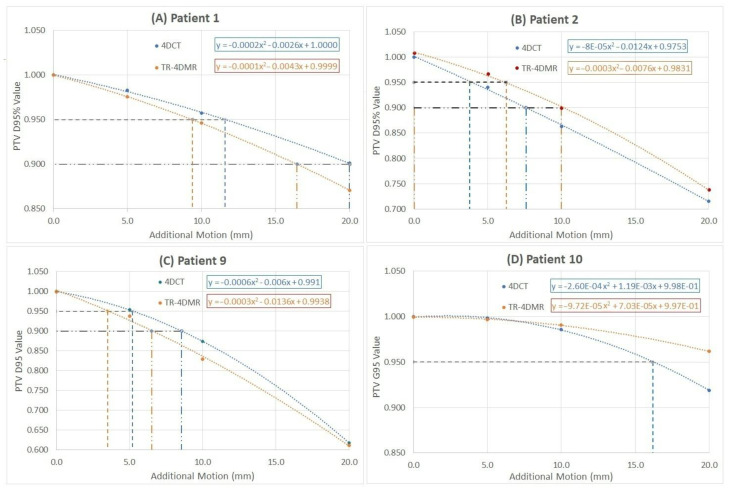
Four examples of the breaching points (PTV D95%) as a function of increased tumor motion amplitudes with quadratic fittings with the equations. Both 4DCT-based (blue) and TR-4DMRI-based (orange) breaching points for patients 1 (**A**), 2 (**B**), 9 (**C**), and 10 (**D**) are shown. For patient 10, only the first breaching point on the 4DCT set can be found.

**Figure 4 jcm-11-07390-f004:**
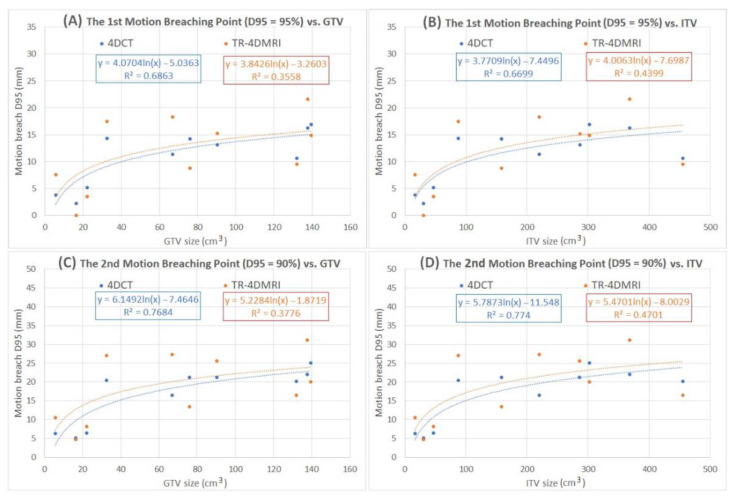
The first and second PTV dose-coverage breaching points (D95% = 95% in the top row and D95% = 90% in the bottom row) as a function of GTV (**A**,**C**) and ITV (**B**,**D**). The logarithm fittings are better than linear fittings (not shown), suggesting that the breaching points are more sensitive to motion amplitude increase for smaller tumors than larger tumors. The “critical mass” of the change is around 25–30 cm^3^ for GTV and 80–100 cm^3^ for ITV. The 4DCT-based results (blue) produce better fitting with higher R^2^ than the TR-4DMRI-based results (orange), because of higher motion variations in multi-breath waveforms than single-breath waveforms.

**Table 1 jcm-11-07390-t001:** Tumor locations, volumes, stages, and native tumor motion amplitudes in 4DCT and TR-4DMRI. Patients 2, 8, and 9 received hypo-fractional stereotactic body radiotherapy (SBRT with a CTV margin of 2 mm) while others received conventional fractionation radiotherapy (with a CTV margin of 7 mm). The volumetric-modulated arc therapy (VMAT) technique is applied in both cases.

Patient	Sex	Age	Site ^#^	Stage	Tumor Volume (cm^3^)			Tumor Motion (mm)	
					GTV	ITV	PTV	4DCT	TR-4DMRI
1	m	59	RLL	IIIa	132	455	705	7.8	10.7
2	f	74	RLL	Ia	6	17	39	8.5	10.8
3	f	77	LUL	IIa	33	87	166	1.4	1.6
4	f	42	LUL	IIa	76	158	281	0.5	0.7
5	m	78	RML	IIa	139	302	447	1.3	2.2
6	f	76	RLL	IIIa	67	220	364	2.4	4.3
7	m	60	RLL	IIIa	90	286	454	3.8	5.6
8	f	75	RLL	Ia	17	30	140	4.0	5.7
9	m	89	RUL	Ia	22	47	98	2.5	4.4
10	f	63	RML	IIIa	138	368	547	0.0	0.1
Average		70			72	197	324	3.2	4.6
St Dev		14			52	153	216	2.9	3.8

^#^ Lung location abbreviation: RLL—right lower lobe, LUL—left upper lobe, RML—right middle lobe, RUL—right upper lobe.

**Table 2 jcm-11-07390-t002:** Clinical ITV-based plans (D95% = 100%) and motion-incorporated plans based on the native and scaled-up motion waveforms derived from 4DCT and TR-4DMRI scans. The actual PTV D95% decreases with the added motion amplitudes using either 4DCT or TR-4DMRI waveforms and the decreasing rate is patient-specific. The motion-incorporated plans with the native tumor motions serve as control experiments, demonstrating the equivalency to the clinical plans. Patients 2, 8, and 9 were treated with hypo-fractional SBRT plans.

Patient	Rx (cGy) Dose × fx	PTV D95% on 4DCT Motion				PTV D95% on TR-4DMRI Motion			
		0 mm	5 mm	10 mm	20 mm	0 mm	5 mm	10 mm	20 mm
1	200 × 30	1.000	0.982	0.957	0.901	0.998	0.976	0.946	0.871
2	1200 × 4 ^#^	0.974	0.915	0.841	0.697	0.982	0.942	0.876	0.719
3	200 × 33	0.999	0.989	0.977	0.904	0.999	0.990	0.979	0.939
4	400 × 15	1.007	1.000	0.978	0.915	1.005	0.988	0.944	0.805
5	400 × 15	1.003	0.994	0.983	0.936	0.997	0.982	0.988	0.900
6	200 × 30	0.997	0.982	0.960	0.853	0.997	0.993	0.980	0.942
7	200 × 33	0.990	0.975	0.957	0.900	0.992	0.975	0.960	0.932
8	1000 × 5 ^#^	0.985	0.875	0.837	0.776	0.925	0.884	0.857	0.797
9	1000 × 5 ^#^	0.991	0.945	0.866	0.612	0.990	0.937	0.829	0.612
10	200 × 30	0.998	0.996	0.984	0.917	0.998	0.995	0.989	0.960
Average		0.994	0.965	0.934	0.841	0.989	0.966	0.935	0.848
St Dev		0.010	0.041	0.061	0.110	0.023	0.035	0.059	0.114

^#^ The prescriptions (Rx) for hypo-fractional stereotactic body radiotherapy (SBRT) treatment. The rest patients were treated with conventionally fractionated radiotherapy.

**Table 3 jcm-11-07390-t003:** The two PTV D95% breaching points (D95% = 95% and D95% = 90%) with increased motion amplitude (in mm) are shown based on quadratic fittings (with R^2^ values). For small tumors (Patients 2, 8, and 9 with GTV < 25 cm^3^), only 2–5 mm and 4–8 mm can lead D95% decrease to 95% and 90%, respectively. For larger tumors (patients 1, 3, 4, 5, 6, 7, and 10 with GTV > 25 cm^3^), however, 9–17 mm and 16–31 mm additional motions are needed to decrease D95% to 95% and 90%, respectively.

Patient	Hypo-Fractional SBRT ^#^: Motion for PTV D95% Breaching Points (mm)						Conventional Fractionation ^%^: Motion for PTV D95% Breaching Points (mm)					
	4DCT			TR-4DMRI			4DCT			TR-4DMRI		
	95%	90%	R^2^	95%	90%	R^2^	95%	90%	R^2^	95%	90%	R^2^
1	-	-	-	-	-	-	10.6	20.2	0.999	9.5	16.5	1.000
2	3.8	6.3	0.999	7.6	10.5	0.999	-	-	-	-	-	-
3	-	-	-	-	-	-	14.3	20.4	0.997	17.5	27.1	0.998
4	-	-	-	-	-	-	14.2	21.2	0.999	8.8	13.4	1.000
5	-	-	-	-	-	-	16.9	25.1	0.948	14.9	20.1	0.966
6	-	-	-	-	-	-	11.4	16.5	0.999	18.3	27.3	1.000
7	-	-	-	-	-	-	13.1	21.3	1.000	15.2	25.6	1.000
8	2.2	5.1	0.985	0.0	4.8	1.000	-	-	-	-	-	-
9	5.2	6.5	0.999	3.5	8.2	0.998	-	-	-	-	-	-
10	-	-	-	-	-	-	16.3	22.0	1.000	21.6	31.2	0.999
Average	3.7	6.0	0.994	3.7	7.8	0.999	13.8	21.0	0.993	14.7	21.7	0.996
St Dev	1.5	0.8	0.008	3.8	29	0.001	2.3	2.6	0.016	4.5	6.5	0.011

^#^ A 2 mm margin is used for early-stage lung cancer to include the microscopic tumor extension for SBRT treatment. ^%^ A 7 mm margin is used for late-stage lung cancer for conventional treatment.

## Data Availability

The patient data used in this study can be available after anonymization upon request.

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
