# Peer review of "A Simulation Study of Tolerance of Breathing Amplitude Variations in Radiotherapy of Lung Cancer Using 4DCT and Time-Resolved 4DMRI"

_jcm, 2022, doi:10.3390/jcm11247390_

Round 1

Reviewer 1 Report

General comments:

It is a very interesting study, and the manuscript ought to be published. However, the manuscript is not particularly well-written, especially there is a mix of past and present tense. This needs to improved and the description of the study has to be clarified further. Also make sure to introduce abbreviations first time you use them

Specific comments:

L12: “lung cancer, affecting” -> “lung cancer patients, and thereby affect”

L15: “(PTV D95%=100%)” consider skipping, this info does not belong here. If kept the abbreviation “PTV” needs to be introduced (if skipped this abbreviation needs to be introduced further down)

L15: “2.0cm” -> “2.0 cm” In general in the whole text, a space is needed between a number and the unit

L19: “repeated/concatenated” are both words needed?

L25: “motion-incorporated plans is 99.4%±1.0% using 4DCT. Tumor motion irregularities are” -> “motion-incorporated plans was 99.4%±1.0% using 4DCT. Tumor motion irregularities were”. In general in the whole text, try to avoid mixing past and present tense.

L29: “breaching points are tumor-size dependent, caused by tumor motion increase” didn’t you define “breaching points” exactly as the size of the increase in tumor movement where the PTV D95% falls below the constraint? Then it does not make sense to write “caused by tumor motion increase”, consider skipping

L29-31: “Clinically, it is important to monitor and avoid baseline drift (systematic) and large motion spikes (random) through threshold-based beam gating.” I don’t think this is supported by your study.

L62: “treatments and found” -> “treatments it was found”

L68: “accounts for 46% of 20-minute scans” I assume this should be “accounts for 46% of the area registered in 20-minute scans” or something similar

L79: “variance of ITV overlap”, overlap with what?

L80: “a strong dependency on native tumor motion range” I am not sure I understand what is meant by this? And how this makes the PDF irreproducible?

L84: “TPS” has this abbreviation been introduced?

L84: “user-developed” potentially “in-house developed”

L85: “the isocenter motion profile while keeping the clinical beam” does isocenter here refer to the tumor movement? Or is it the beam movement?

L89: “at clinically acceptable accuracy” how was this defined?

L94-95: “the equivalency of the motion-incorporated plans based on 4DCT to the ITV-based VMAT plans was first assessed” What is meant here?

L107: “CT/4DCT”, does this mean that some patients only had 3DCT while others had 4DCT? Or how should this be understood?

L114: “(0.75)” what does this number refer to?

L118: “The same FOV” potentially also state the actual size of the FOV

122-123: How was the GTV defined on the different MR images? Did a clinician delineate on each of the scans? Or was rigid or deformable image registration used? Or?

L124-125: “The tumor motion waveforms were confirmed using the diaphragm motion extraction and scaled down to the tumor motion range.” More information is needed here

L126-127: “linearly enlarged by the extra of 5, 10, and 20mm in the motion ranges.” More information is needed here

L130: “4DCT motion simulation” what is meant here? Should it simply have been “4DCT images” / “4DCT scans”?

L149-150: “displaced based on the tumor motion waveforms in sync with the original MLC motion and gantry rotation mimicking VMAT plan delivery” Potentially describe in a bit more detail what is meant by “in sync”

L154: “It is worthwhile to note” Potentially also mention that you are moving the beam but the tumor stays fixed instead of the other way around (if I understand it correctly)

Section 2.4: This can be explained better

Figure 1: What does “Composite Motion-Incorporated Plans” mean? And how was the normalization performed “(2) Normalize CMI plans using the control factor”. These two points need to be explained in the text

L195: “move fewer motions” -> “have a smaller tumor movement”

L197: “hypo-fractional” -> potentially “hypo-fractionated”?

Table 1 potentially belong to the MM section rather than to the Results section

Table 1: Patient 10 has a movement of 0 mm, it is hard to justify why this patient is included. In general, you need to include in the Discussion that these patients all had a very limited tumor motion (only patient 1 and 2 had noticeable tumor movement) therefore to expect that a patient went from having a tumor movement of 0 or 1 mm to have a tumor movement of 20/21 mm is probably not realistic

L221-223: Sentence is missing a verb

L224-225: “The minor difference in PTV D95% for a patient is used to correct the motion-incorporated plans for further comparison.” Which extra correction are you referring to?

L282-283: “evidenced by the logarithm fitting performing much better than the linear fitting and the trend can be visualized in Figure 4.” This seems like a bit of an overstatement considering the low R^2 values even for the logarithmic fitting

Table 3: What are these R^2 values representing?

L322: “others are larger” potentially “others have larger tumor volume”

L334: “motion variations” did you actually look at motion variation or did you assume that the amplitude was the same over the whole course of treatment, and just different from the planning 4DCT (I assume the latter to be the case).

L347: “enlarged tumor motion variations” as mentioned in the previous comment, here you might have to instead write “enlarged tumor motion”

L353: “motions” -> “motion”

L367-368: “the smaller tumor motion in this patient pool should not alter the conclusion of this study.” Except from the thing I also commented above, namely that it is unlikely that the amplitude would increase from 0 mm to 20 mm. This sentence may therefore not be justifiable

L368-370: “For small tumors, when additional motion is added, the ITV size increases are substantial relative to their GTV size due to the cubical relationship between the tumor diameter and the volume of the tumor motion envelope.” Do you here mean when you artificially increase the amplitude? Or if the original amplitude is larger? In the first case, I think this is not what you did, since I assume that you did not actually increase the ITV when you artificially increased the amplitude, or did you? If the latter is correct, then I think the sentence could be changed to: “For small tumors, the ITV volume is substantial larger than the GTV volume for a large tumor motion due to the cubical relationship between the tumor diameter and the volume of the tumor motion envelope.” Or something like this

L372-373: “cause a relatively smaller ITV increase, and therefore, has a smaller impact to the plan quality (PTV D95%).” I start to think you may actually have increased the delineated ITV when you increased the amplitude? Is this correct? If yes, you need to state it in the MM section. But I think this is not the correct way to do it, since the ITV is the way the patient is planned, and if the amplitude is larger than what was planned and you would not notice, then the ITV contour would also not be enlarged.

L375: “control plan” what does this refer to?

L377-379: “This small difference was corrected for the new motion-incorporated plan accordingly when additional motion is added.” 1) a word is missing in this sentence; 2) as mentioned before, you did not describe in the MM section, how this correction was performed

L381-382: “the conclusion drawn here would have a generic impact regardless of native tumor motion.” I am not sure what you base this on, or if I think it is a fair statement

L401: “4.4. Clinical motion management and recommendations” It is probably a bit far fetched to bring any recommendations based on this study

L403: “0-5 mm” If a 0 mm amplitude increase would lead to a dose degradation this must be because of something else than motion. That is, do not state 0 mm here.

L405: “VAMT” -> “VMAT”

L407-408: “which may have lost some of the statistical power of conventional fractionation.” What do you mean here? Do you mean that you only had 3 patients?

Discussion: Almost no references are cited in the Discussion. You need to put your work in relation to other literature on this topic.

L432-433: “Clinically, it is important to monitor and avoid baseline drift (systematic) and large motion spikes (random) through threshold-based beam gating” I don’t think this is supported by your study.

Author Response

Please see the attached file with the point-to-point response.  Thanks.

Reviewer 2 Report

The authors presents a study evaluating the dosimetric impact of the motion of pulmonary mobile lesions under modern conditions of management including the use of 4DCT.

They are based on a literature which already underlines the limits of the definition of the ITV from the 4DCT based on the experience acquired with cyberknife tracking, fluoroscopy and 4DMRI.

They assess the impact of size on this loss of coverage induced by unplanned movements.

The methodology is well built based on a 4DMRI to establish a DVF as well as an Eclipse plug-in to further  increase the tumor motion as tumor motion waveforms initially established from the 4DCT and the 4DMRI.

The 10 cases were treated for lung carcinomas.

The study is interesting especially when it assess quantitatively the link between the loss of dose and the amplitude of motion and its variation with tumor volume.

A missing element is in my opinion the information concerning the algorithm used, which can have an impact on the evaluation of the penumbra.

Finally, contrary to the elements of the discussion, I do not think that we can extend as is these results shown with bronchial tumors to other mobile tumors: liver, pancreas.

Author Response

Please see the attached file with the response.  Thanks.

Round 2

Reviewer 1 Report

General comments:

The mix of past and present tense still needs to fixed.

Specific comments:

L18: “(PTV D95% = 100%)” -> “(PTV D95%)”

L21: “represents” -> “represented”

L25: “in motion-incorporated” -> “in the motion-incorporated”

L26 + rest of manuscript: “4mm” -> “4 mm”

L28-29: “This study has demonstrated that PTV D95% breaching points may occur during treatment delivery” No it did not. This study only showed that the breaching points may occur IF tumor movement variations of a certain size occurs, but since you did not look into if such large tumor movement variations actually occurs, you need to rephrase this sentence

L49-54: “… even during treatment, and therefore, the ITV delineated…” I guess there in between these two statements needs an explanation that the discomfort of the patient can lead to changes in the breathing pattern, which therefore results in an incorrect ITV

L67: “2D cine” should this be “2D cine MR imaging”?

L79: “a high variance of ITV overlap among 10-second scans” this sentence is still not understandable in the context

L92-93: “and simulated based on” consider skipping

L109: “clinical CT-4DCT simulation” this still does not make sense. If you use the 3D image in your study then rephrase to “clinical CT simulation, acquiring both a regular 3D CT and a 4DCT, both in free breathing,”. If you don’t use the 3DCT in this study then just rephrase to “clinical 4DCT simulation”

L121: Consider “FB image acquisitions” -> “FB MR image acquisitions”

L129: Consider “the two motions” -> “the diaphragm motion and tumor motion”

L135: “image scans” -> “images”

L141-142: Consider “the PTV has a 5.0 mm setup margin over the ITV” -> “the PTV was a 5.0 mm isotropic expansion of the ITV”

L161: “lung cancer” -> “lung cancer patients”

L200: “dotted” -> “dashed”

Table 1 (and rest of manuscript): Consider ordering the patients so the 3 SBRT patients are patient 1-3, or 8-10. It would make the description easier. You could furthermore consider to sort the patient based on their tumor volume or tumor motion

L238: “0.994±0.010” -> “99.4%±1.0%”

L240-241: “for a patient in the control is” -> “between the clinical and the control plan was”

Fig 2: Too low image quality. The text is not readable. This needs to be improved.

L281: Consider “the nonlinear fits” -> “the quadratic fits”

L294: “of GTV and ITV” -> “of GTV and ITV volume”

L340: “VAMT” -> “VMAT”. Please read your manuscript through before submitting

L353: Consider “value, or the” -> “value, denoted the”

L358-359: “then multi-fold large tumor motions on average” rephrase

L439-440: Consider “which have 5 fractions or fewer and therefore lost the statistical averaging power in treatment delivery comparing with conventional fractionation” -> “which have 5 fractions or fewer, whereby the averaging effect of having multiple fractions is lost in comparison with conventional fractionation”

L440-442: This sentence needs a reference. Not being worried since no data is available seems unlogical. Therefore, you need to bring a reference saying that we don’t need to worry.

L449: “more reliable representation” this statement needs a reference

Author Response

Please see the attached response word file.  Thanks.
